# Genomic Analysis Highlights Putative Defective Susceptibility Genes in Tomato Germplasm

**DOI:** 10.3390/plants12122289

**Published:** 2023-06-12

**Authors:** Ruiling Li, Alex Maioli, Sergio Lanteri, Andrea Moglia, Yuling Bai, Alberto Acquadro

**Affiliations:** 1Plant Genetics and Breeding, Department of Agricultural, Forest and Food Science (DISAFA), University of Torino, 10095 Grugliasco, Italy; ruiling.li@unito.it (R.L.); alex.maioli@unito.it (A.M.); sergio.lanteri@unito.it (S.L.); andrea.moglia@unito.it (A.M.); 2Plant Breeding, Wageningen University & Research, 6708 PB Wageningen, The Netherlands; bai.yuling@wur.nl

**Keywords:** tomato germplasm, genome sequencing, susceptibility genes, SNPs, CRISPR/Cas9

## Abstract

Tomato (*Solanum lycopersicum* L.) is one of the most widely grown vegetables in the world and is impacted by many diseases which cause yield reduction or even crop failure. Breeding for disease resistance is thus a key objective in tomato improvement. Since disease arises from a compatible interaction between a plant and a pathogen, a mutation which alters a plant susceptibility (S) gene facilitating compatibility may induce broad-spectrum and durable plant resistance. Here, we report on a genome-wide analysis of a set of 360 tomato genotypes, with the goal of identifying defective S-gene alleles as a potential source for the breeding of resistance. A set of 125 gene homologs of 10 S-genes (*PMR 4*, *PMR5*, *PMR6*, *MLO*, *BIK1*, *DMR1*, *DMR6*, *DND1*, *CPR5*, and *SR1*) were analyzed. Their genomic sequences were examined and SNPs/indels were annotated using the SNPeff pipeline. A total of 54,000 SNPs/indels were identified, among which 1300 were estimated to have a moderate impact (non-synonymous variants), while 120 were estimated to have a high impact (e.g., missense/nonsense/frameshift variants). The latter were then analyzed for their effect on gene functionality. A total of 103 genotypes showed one high-impact mutation in at least one of the scouted genes, while in 10 genotypes, more than 4 high-impact mutations in as many genes were detected. A set of 10 SNPs were validated through Sanger sequencing. Three genotypes carrying high-impact homozygous SNPs in S-genes were infected with *Oidium neolycopersici*, and two highlighted a significantly reduced susceptibility to the fungus. The existing mutations fall within the scope of a history of safe use and can be useful to guide risk assessment in evaluating the effect of new genomic techniques.

## 1. Introduction

Tomato (*Solanum lycopersicum* L.) is one of the most widely grown vegetables in the world. The species is subjected to many diseases, which cause substantial economic losses. Breeding for disease resistance is thus a key objective in tomato improvement. Breeders’ efforts have been mainly focused on the introgression of resistance genes (R-genes) in elite genotypes, a strategy which is time consuming and often not durable [1], however, as most resistance genes confer race-specific resistance and are frequently overcome by a pathogen’s new virulent race.

Most pathogens require the cooperation of the host to establish a compatible interaction, which is mediated by host susceptibility (S) genes [2,3,4]. The identification of S-genes’ spontaneous mutants represents an emerging breeding strategy for durable and broad-spectrum resistance [5]. A mutated S-gene named Mildew Locus O (*MLO*) was found to induce resistance to *Blumeria graminis* f. sp. *hordei* in barley and has been used for over 70 years in breeding programs [6]. Many *MLO* orthologs have been identified in several monocots and eudicots species [7], including tomato, and it was found that mutations in *MLO* confer resistance to powdery mildew as demonstrated with various mutagenesis approaches including chemical mutagenesis, RNAi, and CRISPR-Cas9 [8,9,10,11,12]. Homoeoalleles in hexaploid bread wheat have also been modified using transcription activator-like effector nuclease (TALEN) and CRISPR-Cas9 to confer heritable resistance to powdery mildew [13]. Mutations in the *PMR4* (powdery-mildew-resistant) gene have also been found to induce not only resistance to powdery mildew but also late blight [14]. Its CRISPR/Cas9-based disabling reduced susceptibility to both pathogens in tomato [15,16] and in potato to late blight as well as several diseases [17]. *PMR5*, a pectin acetyltransferase, and *PMR6,* a pectate-lyase-like gene, increased powdery mildew resistance in their *Arabidopsis* mutants, despite the mutants showing higher susceptibility to multiple strains of *Botrytis cinerea* [18,19,20]. In *Arabidopsis,* mutation of the *BIK1* (Botrytis-induced kinase1) gene, which belongs to the family of RLCKs, has been found to play a role in defense against pathogens and insects acting specifically or redundantly in immune signaling [21], and it induced strong resistance to *Plasmodiophora brassicae* [22], although it also increased the susceptibility to green peach aphids [23].

The *DND1* (Defense No Death 1) gene is a cyclic nucleotide-gated ion channel protein, and *Arabidopsis* mutants showed broad-spectrum resistance against several fungal, bacterial, and viral pathogens due to disturbance of the Ca^2+^/calmodulin-dependent signaling pathway [24]. In potato and tomato, RNAi silencing of *DND1* orthologs led to resistance to late blight and to two powdery mildew species (*Oidium neolycopersici* and *Golovinomyces orontii*) [17,25].

Other genes have been observed to play important roles in regulating plant defense mechanisms against different pathogens in plants (*DMR1*, *DMR6*, *CPR1-1*, *CPR5-2*, and *SR1*). In *Arabidopsis*, the *DMR1* gene (coding homoserine kinase) mediates susceptibility mechanisms occurring in both vegetative and reproductive plant tissues during infection by both obligate biotrophic oomycete and hemi-biotrophic fungal pathogens. Its mutation conferred enhanced resistance to *Fusarium graminearum* and *F. culmorum*, which cause ear blight disease in cereals [26]. Recently, the ortholog of *AtDMR1* was efficiently disabled through the CRISPR/Cas9-mediated editing system in *Ocimum basilicum*, but no results on its effect on pathogen resistance were reported by Navet and Tian (2020) [27]. In *Arabidopsis*, the resistance to Cauliflower mosaic virus (CaMV) is regulated by the salicylic acid (SA) and jasmonic acid/ethylene (JA/ET) signaling pathways. Mutations in the constitutive expressor of PR genes *(CPR1*-*1* and *CPR5*-*2*)*,* involved in their biosynthetic pathway, resulted in constitutive activation of SA-dependent defense signaling and increased resistance to systemic infection of CaMV [28]. Furthermore, the *cpr* mutants, including *cpr5*, exhibited both *EDS*-1-dependent and -independent components of plant disease resistance [29]. The *SR1* (signal responsive1) gene is a calmodulin-binding transcription factor, modulating plant defense. A gain-of-function mutation in *SR1* by gene editing in *Arabidopsis* enhanced disease resistance to powdery mildew and regulated ET-induced senescence by directly regulating non-race-specific disease resistance1 (*NDR1*) and ethylene insensitive3 (*EIN3*) genes [30]. Similarly, in tobacco, *sr1* mutants (*ater1* to *ater7*) generated via T-DNA activation tagging were less susceptible to tobacco mosaic virus due to reduced microtubule dynamics [31]. One of the most intriguing S-genes is DOWNY MILDEW RESISTANCE 6 (*DMR6*), encoding for a 2-oxoglutarate (2OG)- and Fe(II)-dependent oxygenase, which has salicylic acid (SA) 5-hydroxylase activity and thus reduces the active SA pool [32]. Inactivation of DMR6 results in increased SA acid levels [33,34]. Tomato *sldmr6-1* mutants, characterized by high accumulation of SA, showed enhanced resistance against evolutionarily distinct classes of pathogens: bacteria, oomycetes, and fungi [34].

The EFSA (European Food Safety Authority) has recently released scientific opinions on plants obtained through new genomic techniques, i.e., targeted mutagenesis based on gene editing, cisgenesis, and intragenesis, and elaborated criteria for the risk assessment of plants obtained through new genomic techniques [35]. EFSA proposed six main criteria to assess risk assessment [36] among which was the history of safe use. If familiarity and/or history of safe use can be demonstrated as a result of traditional usage and/or widespread cultivation, the donor plant and/or gene/allele and the associated trait can be subjected to a reduced risk assessment [37]. In other words, the risk assessment will consider both the probability for such an allele to be obtained by conventional breeding or to be already in place in the breeders’ gene pool. A genomic survey on the genetic diversity already present in the germplasm of a species can assist this step. The knowledge of existing defective alleles in the germplasm of a species can assist this risk assessment step and provide a resource for tomato genomic-assisted breeding programs as well as tailored gene editing approaches for resistance to biotic stresses.

Here, we analyzed in a set of 360 resequenced tomato genotypes, a set of 125 genes belonging to ten S-gene families (*PMR4*, *PMR5*, *PMR6*, *MLO*, *BIK1*, *DMR1*, *DMR6*, *DND1*, *CPR5*, and *SR1*), with the goal of evaluating the frequency of high-impact mutations and highlight potential sources of broad-spectrum and recessively inherited resistance. The identified mutations were screened to assess their likely impact on protein functionality. Genotypes carrying high-impact homozygous SNPs in S-genes were assayed for resistance to *O. neolycopersici*, of which two highlighted reduced susceptibility. Moreover, sgRNA sequences were designed for eight S-genes, and they were made available for the creation of optimal gene editing constructs.

## 2. Results and Discussion

In order to identify natural mutant alleles of tomato S-genes, we analyzed the genomic diversity of the cultivated tomato germplasm consisting of a set of 360 genotypes (Appendix A). The data were divided into different datasets: (1) a collection of 168 big-fruited *S. lycopersicum* accessions (fruit weight = 111.33 ± 68.19) and 17 modern commercial hybrids (F1), altogether called BIG); (2) a collection of 53 *S. pimpinellifolium* accessions (fruit weight = 2.04 ± 0.85 g, called PIM); (3) a collection of 112 *S. lycopersicum* var. *cerasiforme* accessions (fruit weight = 13.29 ± 9.54, called CER). The whole collection of 360 genotypes was referred to as ALL. We selected 10 S-genes (Appendix A), of which some are known to reduce susceptibility to pathogens when knocked out or knocked down [2]. The selected S-genes including *PMR4*, *PMR5*, *PMR6*, *MLO*, *BIK1*, *DMR1*, *DMR6*, *DND1*, *CPR5*, and *SR1*, which facilitate host compatibility by being involved in host recognition and penetration, negative regulation of host immunity, or pathogen proliferation. This work represents the first examination at a genomic level of S-genes and existing putative defective alleles in the *Solanaceae* family.

Initially, a blastP analysis was performed (Figure 1) to identify homologs from the 10 chosen genes. A total of 125 S-gene homologs were obtained and used for further analyses (Table 1). The genome sequences of 360 accessions [38] were analyzed (Appendix A, genotypes) for SNP mining, and 11,620,517 SNPs/indels were detected across 34,725 tomato gene locations. The number of SNPs over 185 accessions (BIG) was 7,744,233 (67%). In the 125 gene member subset (Table 1), 54,000 SNPs/indels were observed using the SNPeff pipeline. Among these, 51,000 had no effect on protein function, with them being synonymous with SNPs or located in intergenic regions. A total of 1500 SNPs had a low impact, and 1300 had a moderate impact. A total of 119 high-impact SNPs were observed. The distribution of these SNPs was studied among the 10 S-genes (Figure 2).

Despite differences in the number and type of genes considered, recent analyses on the nucleotide diversity of S-genes in other species such as apple [39,40] and grape [41] have been conducted. The number and density of SNPs observed in grape (*V. vinifera*) was ~15 SNPs per Kb (1SNP every 66 bp), while in both wild species and hybrid/wild Vitis species, it was 18 SNPs per Kb (1 SNP every 55 bp) [41]; in apple (*M. domestica),* in *Mlo-like* genes, values of ~41 SNPs per Kb and 1SNP every 24 bp were observed [39]. These values were higher than the ones we obtained, i.e., 1 SNP every 1031 bp in the whole dataset and 1 SNP every 472 bp in tomato (BIG), reflecting the different genetic structures of the species, the homozygosity level, and their domestication history.

Our analyses (Table 1) showed that when both wild and cultivated tomato genotypes were considered, the number of SNPs and their density were higher (119 SNPs with a density of 1 SNP per gene). However, when only “big tomato” genotypes were considered, the number of SNPs and their density was halved (58 SNPs with a density of 0.5 SNPs per gene); this suggests that there is a specific reservoir of S-gene alleles in the wild tomato germplasm that can be used for breeding. Haplotype analysis of the 119 SNPs was conducted, revealing the presence of specific conserved haplotypes (Appendix A) that were clearly distinguishable from other haplotypes, providing useful information for breeders.

We analyzed the potential impact of 119 highly detrimental mutations, including frameshift-inducing mutations that result in major damage such as knock-out mutations. However, there are also many moderate-impact mutations (1326) that may lead to changes in protein conformation and function. Although we did not delve into these effects in detail, they are worth monitoring in order to gain a deeper understanding of altered S-genes. Among the 119 SNPs, 10 were validated in 10 genotypes readily available within the research group facilities (http://eurisco.ecpgr.org, accessed on 1 April 2022) through Sanger sequencing with a 90% validation rate (Appendix A); indeed, some non-validated SNPs were mutations detected in a heterozygous condition or possessed the same allelic profile as the reference; the emergence of such heterozygous/reference-like SNPs during the validation step can be explained by the high genetic diversity existing within the analyzed germplasm set (Figure 3), as observed by Li et al. [15].

The number of SNPs in each family was related to their length, but the SNP density appeared higher in certain genes (Table 1, *PMR 4*, *PMR5*, *PMR6*, *MLO*, *BIK1*, and *CPR5*) and lower in others (*DMR1*, *DND1*, *SR1*, and *DMR6*). This difference might be due to the fact that some genes are single-copy genes or present in a nodal position (hub) within the cell regulation network, hardly supporting deleterious SNPs [42]. On the contrary, the presence of multiple genes in a gene family may mitigate the impact of deleterious mutations [43]. In specific cases, such as *DMR1*, a single-copy tomato gene exhibited a deleterious mutation (a gained stop codon) in homozygosity, but its potential impact on protein functionality was likely reduced, as the causative SNP was located in the last six codons of the gene (1129/1134) (Appendix A). In some others (e.g., *BIK1*-like genes), many occurrences were observed since all the 51 serine-threonine kinases, belonging to the RLCK (clade VII) repertoire, were analyzed.

Based on the nucleotide diversity (Pi) analysis, we observed that bottleneck events appear to be present in some S-genes (Appendix A, *BIK*-Solyc01g028830 and Solyc05g007050; *DMR1*-Solyc04g008760; *DND*-Solyc02g088560; *MLO*-Solyc06g082820; *PMR4*-Solyc01g006350; and *PMR6*-Solyc06g071020) considering BIG varieties in relation to the other two groups (PIM and CER). In general, we found that the PIM group showed the highest diversity in S alleles, while the CER group exhibited a moderate level of diversity, and the BIG group showed the least diversity in accordance with the known tomato history of domestication. Indeed, in few cases the genetic variation is reduced in BIG and CER in a similar way while not in PIM (e.g., *MLO*-Solyc02g077570 and *PMR4*-Solyc01g006350); in others (e.g., *BIK*-Solyc06g005500; *BIK*-Solyc06g083500; and *PMR6*-Solyc06g071020), the genetic variation is reduced in all the three groups.

### 2.1. Homozygous SNPs/Indels

The number of genotypes with two SNPs was 174 (whole dataset) and 36 (BIG tomatoes), while those with three or more SNPs were 114 and 14 (Table 2, Figure 4), respectively. This high representation can be explained by the presence of multigene families such as *BIK1*-like which might present some degree of redundancy. While examining those high-impact mutations, the results revealed that certain mutations appeared frequently in the cultivated germplasm and were preserved across various genotypes, as displayed in Appendix A. One example is *BIK1* (Solyc05g024290, SNP in chr5:31013858), which could be maintained under selective pressure in clustering genotypes within the germplasm materials (Figure 3, e.g., Rowpac, M-82, Santa Chiara, Hunt101, Puno I, and E-6203). The genotypes carrying a high number of SNPs (three or more) were approximately a dozen (e.g., Panama, N 739, Rowpac, Micro-Tom, Guayaquil, Droplet, M-82, Hawaii 7998, and KR2), and information about these SNPs is provided in Appendix A. Certain mutations, such as *BIK1*-like/Solyc01g008860 and *DMR1*-like/Solyc04g008760 in specific genotypes (e.g., N-739/TS-074), appeared to be of lower relevance as they were present in the final percentile of the sequence length (Appendix A).

### 2.2. Heterozygous SNPs/Indels

The incidence of deleterious SNPs in S-genes in a heterozygous condition was comparatively lesser than that of homozygous ones, as observed in both the complete germplasm collection (ALL) and the BIG tomato varieties (Table 2, Figure 4). This frequency may be due to the genetic structure of tomato as an inbred species, which tends to have a low number of heterozygous mutations [15]. However, the number appears relatively high because such mutations, although harmful, can be maintained in the genome if the normal allelic copy continues to function. This high frequency is particularly noticeable in the case of multiple member S-genes (e.g., *BIK1*-like genes) which may exhibit some redundancy and have no effects, or due to the position of the SNP within the gene (e.g., *DMR1*/Solyc04g008760 in TS-113 and *BIK1*-like/Solyc01g008860 in Chiclayo, Appendix A). If two SNPs are considered, the number of genotypes was 89 (ALL) and 10 (BIG), while if three SNPs are considered, the number of genotypes decreases to 54 (ALL) and 3 (BIG). Some heterozygous mutants for S-genes were also identified, which have a 50% chance of acquiring resistance through natural mutagenic effects.

### 2.3. sgRNA Design

Introgression of S-genes’ alleles through breeding into elite varieties is possible, but it is a long and labor-intensive process and has limitations due to linkage drag. To address this issue, in analogy with the work from Prajapati and Nain [44], sgRNA sequences were designed for eight of the proposed S-genes (Appendix A) and made available to a wider audience through the creation of optimal gene editing constructs. In total, 113 sgRNAs were designed, considering only the highly specific categories (A0, B0, A0.1, and B0.1) for CRISPR-Cas9-mediated genome editing to minimize off-target events. Specifically, 39 A0, 20 A0.1, 48 B0, and 6 B0.1 sgRNAs were designed. Each gene was equipped with at least one useful sgRNA, with *PMR4*, *PMR5*, *PMR6*, *MLO1*, and *BIK1* having the most sgRNAs at 13, 15, 20, 8, and 50, respectively.

### 2.4. Disease Assay

As a preliminary assay, five genotypes, readily available within the research group facilities (http://eurisco.ecpgr.org, accessed on 1 April 2022), were selected for a disease assay to assess their resistance to *O. neolycopersici (On)*. They included three varieties (PunoI/TS-108, Droplet/TS-296, and M82/TS-003) with deleterious SNPs and two varieties with no deleterious SNPs in the S-genes (VF-36/TS-01 and Moneymaker/TS-02). M-82 carried three mutated genes (*BIK1*-*like* Solyc05g024290 and Solyc04g050970, and *PMR4*-like/Solyc01g073750), which introduced a stop codon and produced truncated proteins. Puno-I carried two mutated genes (*BIK1*/Solyc05g024290 and *PMR4*/Solyc01g073750) in the middle of the gene, resulting in truncated proteins. Droplet had four high-impact mutations, including one in the *BIK1-like* gene (Solyc04g050970), two in the *Mlo1-like* gene (Solyc02g077570), and one in the *PMR4-like* gene (Solyc01g073750). These varieties showed sequences that predicted the presence of truncated susceptibility proteins in a homozygous state. To assess whether these selected varieties with deleterious SNPs in S-genes had higher resistance to powdery mildew, we inoculated all of them with *O. neolycopersici* and evaluated the disease index (Table 3, Appendix A). Two of them (Puno1 and M-82) showed reduced susceptibility to *O. neolycopersici* based on visual scoring of disease symptoms, while no significant differences in the disease index were observed in the others. The reason for this incomplete resistance may lie in the genes under consideration (*BIK1-like*: Solyc05g024290 and Solyc04g050970). The RLCK family encodes for a series (~50) of serine/threonine protein kinases with a role in post-translational regulation through, in the case of *BIK-1*, the phosphorylation of *FLS2* and *BAK1* [45,46]. The latter gene is involved in pathogen-associated molecular pattern (PAMP)-triggered immunity (PTI) signaling, including calcium signaling and defense responses downstream of *FLS2*. With the RLCK subfamily VII being a large clade (46 members in Arabidopsis; 51 in the present work), whose members play a role both specifically or redundantly in immune signaling, some *BIK1*-like genes could have a vicarious role in case of the emergence of mutant forms (e.g., Solyc04g050970 (49.186.199 bp, chromosome 4) in M82 and Solyc05g024290 (31.013.858 bp, chromosome 5) in the PunoI and M82 genotypes. Moreover, the *Mlo1-like* (Solyc02g077570) and *PMR4-like* (Solyc01g073750) genes were found to differ from the *SlMlo1* and *PMR4* genes (Table 1), which were previously known to provide complete resistance in the presence of a loss-of-function allele. Our research was an extensive genomic study incorporating a small pilot study on the impact of mutations on pathogenesis. We carried out pathogenesis assays using plant material readily available at our academic institutions. However, restrictions imposed by the recent Nagoya protocol on plant material transfer and difficulties in obtaining material for phytosanitary reasons limited our scope. We propose further research on accessions such as Panama, N739, and Rowpac (which have 6, 5, and 5 homozygous deleterious SNPs respectively)—a poorly characterized plant material that deserves further investigation. These materials should also be analyzed using different fungal pathogens (*Phytophthora infestans*, *Botrytis*, etc.) or bacteria (*Pseudomonas syringeae*).

## 3. Materials and Methods

### 3.1. Data Mining on S-Genes

A preliminary blastP (https://ftp.ncbi.nlm.nih.gov/blast, accessed on 1 March 2022) analysis allowed us to identify the possible orthologs for susceptibility genes, using information from different plant species (from Schie and Takken [2]; Appendix A) and by considering as a preferential choice criterion the e-value (range 0–1 × 10^−10^) and the percentage of similarity and the query coverage. Since many genes were present in multigene families, the filtering criteria varied, and previous functional annotations were used to filter out non appropriate candidates.

### 3.2. SNP/Indel Data

The genotypic data discussed in Lin et al.’s study [38] were retrieved from SGN (ftp://ftp.solgenomics.net/genomes/tomato_360, accessed on 1 April 2022) as raw vcf files. The data derived from 360 genotypes (Appendix A) were divided into different datasets: (1) a collection of 168 big-fruited *S. lycopersicum* accessions and 17 modern commercial hybrids (F_1_), altogether called BIG); (2) a collection of 53 *S. pimpinellifolium* accessions (called PIM); (3) a collection of 112 *S. lycopersicum* var. *cerasiforme* accessions (called CER). the whole collection of 360 genotypes was referred to as ALL. A principal component analysis (PCA) was conducted using the R-based ClustVis suite (https://biit.cs.ut.ee/clustvis, accessed on 1 June 2022). The dataset used for PCA was the whole dataset pruned and filtered using vcftools (https://vcftools.github.io, accessed on 1 June 2022), using the option --max-missing = 0.2, for filtering loci. The genetic variation of the S alleles was analyzed using the nucleotide diversity (Pi) index implemented in vcftools (https://vcftools.github.io, accessed on 1 June 2022) among the different groups (PIM, CER, and BIG). We focused on a 100 kb region, centered around each deleterious SNP with a 5 k window.

### 3.3. SNP Annotation

The SNP data were newly annotated using the v2.5 assembly with ITAG2.4 information. The SnpEff v5.0 program was adopted to infer functional annotation of any SNPs/indels and any potential deleterious effect on protein structure [47]. The effect of each SNP/indel was classified into four of classes of effects: (1) high effect, as variants changing the frameshift, thereby introducing/eliminating stop codons or modifying splice sites; (2) moderate effect, as variants altering the aminoacidic sequence; (3) low effect, as synonymous variants in coding regions; and (4) modifier effect, as variants located outside the coding sequence (non-transcribed regions or introns). Annotated vcf files from each individual were merged into a single file to integrate the entirety of the information. Bedtools intersect (https://github.com/arq5x/bedtools2, accessed on 1 June 2022) was used to screen for overlaps between the genomic features related to the S-genes (in gff format), and the SNP positions emerged from the SnpEff analysis; genomic coordinates were lifted over from SL2.50 to SL5.0 [48]. Functionally annotated SNPs from both the BIG and ALL dataset were inspected for different categories (high, moderate, and low impact) and were considered and counted for each accession, through custom bash scripts. Conserved deleterious SNPs were utilized as informative markers for generating haplotypes of SNPs, and the resulting haplotype information was analyzed around the S-genes using the software tool Tassel. All the categories were decomposed into homozygous and heterozygous SNPs/indels. A subset of SNPs was validated through Sanger sequencing (BMR Genomics Service, Padova, Italy) of the PCR-amplified gene fragments using the primers listed in Appendix A. 

### 3.4. Single Guide RNA (sgRNA) Design on Target Genes

The CRISPR-PLANT v2 platform (http://omap.org/crispr2/CRISPRsearch.html, accessed on 1 July 2022) was used to design sgRNAs in S-genes using the gene code as a query for the scan of the SL2.5 genome. We selected sgRNAs only present in exons, discarding the ones with a high possibility to give off-targets. Then, the rest of the sgRNAs were selected using their quality, based on the mismatch score in their seed sequence. The sgRNAs were divided by the CRISPR-PLANT software into different quality classes (A0, B0, A0.1, B0.1, A1, B1, A2, and B2), with A0 being the most specific and B2 being the least specific. The sgRNA sequence of each selected S-gene and the relative quality is reported in Appendix A; only the A0, A0.1, B0, and B0.1 classes were reported in the output as highly specific sgRNAs for CRISPR-Cas9 mediated genome editing.

### 3.5. Disease Assay

Thirty seeds of selected accession, three with mutations (M-82, Puno-I, and Droplet) and two controls (VF-36 and MoneyMaker) were sowed and then inoculated with the Wageningen University isolate of *O. neolycopersici* (On) by spraying 4-week-old plants with a suspension of conidiospores obtained from the leaves of infected tomato Moneymaker plants and adjusted to a concentration of 3.5 × 10^4^ spores per ml. The Moneymaker variety was used as the susceptible control. The inoculated plants were grown at 20  ±  2  °C with 70  ±  15% relative humidity and a day length of 16 h in a greenhouse of Unifarm of Wageningen University & Research, the Netherlands, and placed randomly within the greenhouse. Disease index scoring was carried out 10 and 12 days after inoculation. Symptoms were scored visually using a scale from 0 to 3 as described by Huibers et al. [14]. Statistical differences between each variety and the control were analyzed using a two-tailed *t*-test (* *p* < 0.05).

## 4. Conclusions

In this study, we conducted a comprehensive genomic survey of various tomato genotypes to identify putative defective alleles of susceptibility genes. Our analysis revealed the presence of natural homozygous/heterozygous deleterious alleles, which we further validated through Sanger sequencing. Interestingly, we observed that certain genotypes carrying high-impact homozygous SNPs in S-genes exhibited significantly reduced susceptibility to *O. neolycopersici*. These findings offer valuable insights for plant genetics and have the potential to enhance genomic-assisted breeding programs focused on developing resistance to biotic stresses. Nonetheless, it is important to acknowledge that incorporating desirable alleles into elite genotypes can be a time-consuming process with challenges such as linkage drag. To address this, we have also explored the application of a gene editing approach using single guide RNA (sgRNA) design. This alternative method shows promise in disabling targeted genes, presenting a powerful means to obtain elite tomato genotypes resistant to biotic stresses. Furthermore, our genomic survey can contribute to the evaluation and risk assessment of new genomic techniques by tracking existing alleles in the context of their “History of Safe Use” [36].

This study underscores the significance of publicly available data in enabling further analyses and, more importantly, highlights the wealth of potentially beneficial alleles already present in the existing tomato breeding pool. If proven to reduce disease susceptibility, these genes could serve as long-lasting sources of tolerance against various pathogens.

## Figures and Tables

**Figure 1 plants-12-02289-f001:**
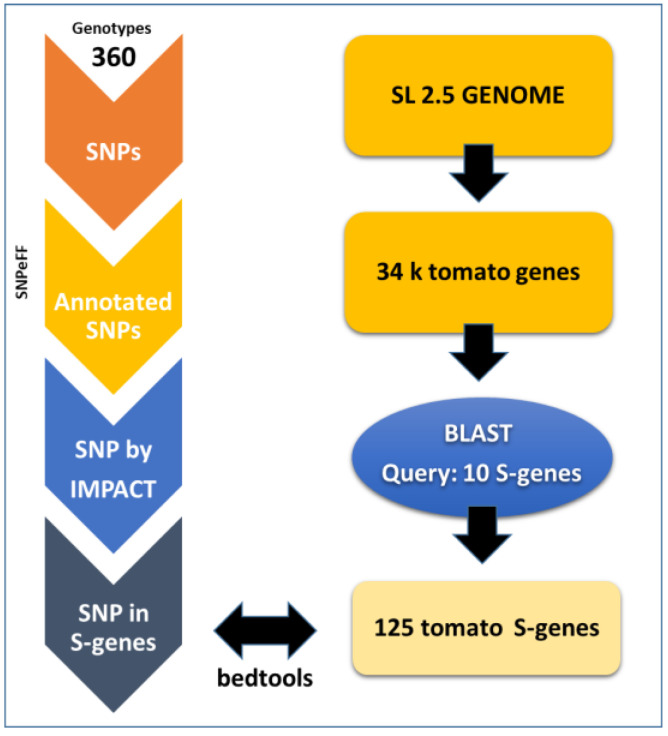
Flowchart of the high-impact SNP mining process within the available sequenced tomato germplasm (the data were originally retrieved from Lin et al. 2014 [38]).

**Figure 2 plants-12-02289-f002:**
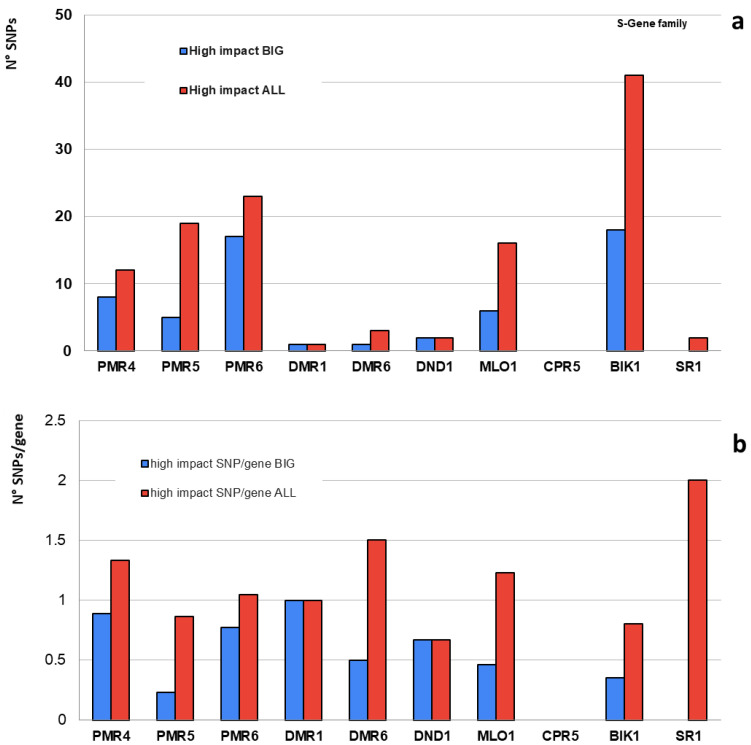
(**a**) Distribution of high-impact SNPs in the S-gene families (N° SNPs); (**b**) relative SNP density (N° SNPs/gene).

**Figure 3 plants-12-02289-f003:**
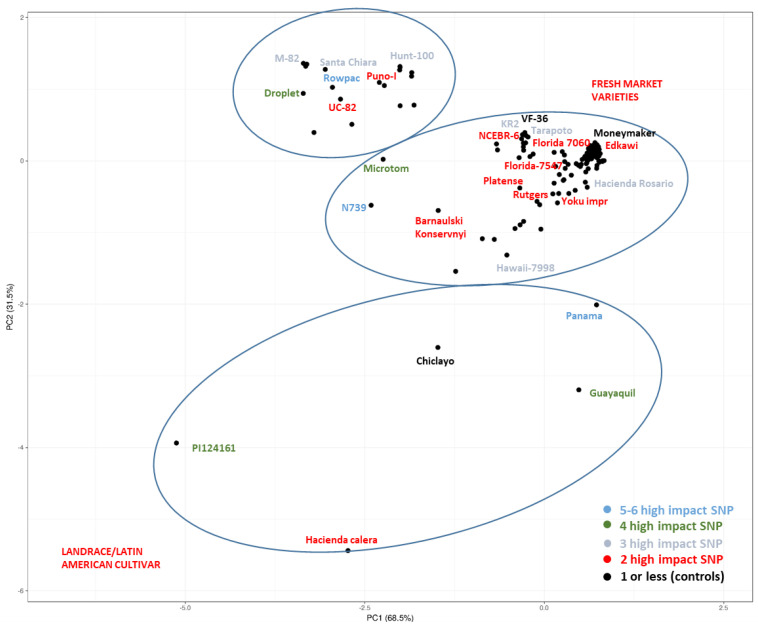
Genotypes accumulating multiple mutations in S-genes. In light blue are the reported genotypes with five or more SNPs, in green are the genotypes with four SNPs, in gray are the genotypes with three SNPs, in red are the genotypes with two SNPs, and in black are the rest of the genotypes (0–1 SNPs).

**Figure 4 plants-12-02289-f004:**
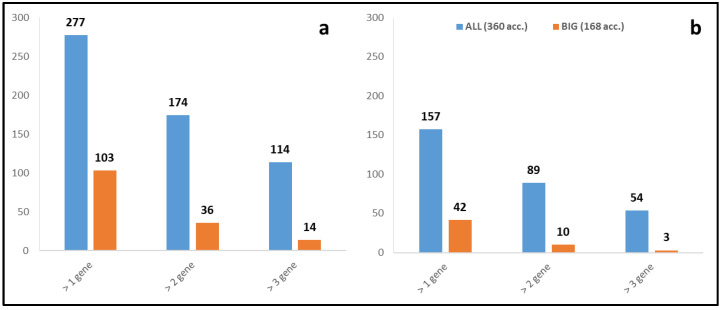
Genotypes accumulating mutations in S-genes in (**a**) homozygous and (**b**) heterozygous states.

**Table 1 plants-12-02289-t001:** Statistics on SNPs/indels within S-genes related to the 360 panel. The numbers are always formed by two values X/Y, where X is the number of SNPs observed in the 360 panel and Y is the number of SNPs observed in the tomato panel. BIG = 168 *S. lycopersicum* + 17 F_1_ hybrid genotypes; ALL = 168 *S. lycopersicum* + 17 F_1_ hybrid genotypes *+* 53 *S. pi +* 112 *S. cerasiforme +* 10 wild tomatoes.

S-Gene Family	Tomato Ortholog	Genes	High Impact		High Impact (SNP/Gene)		Moderate Impact		Low Impact		N° Variants (Total)		Total SNP/Gene	
			BIG	ALL	BIG	ALL	BIG	ALL	BIG	ALL	BIG	ALL	BIG	ALL
*PMR4*	Solyc07g053980	9	8	12	0.9	1.3	95	199	166	288	2473	4033	274.8	448.1
*PMR5*	Solyc06g082070	22	5	19	0.2	0.9	172	274	151	257	3341	5267	151.9	239.4
*PMR6*	Solyc11g008140	22	17	23	0.8	1.0	104	188	120	187	8065	12,989	366.6	590.4
*DMR1*	Solyc04g008760	1	1	1	1.0	1.0	6	6	6	12	147	215	147.0	215.0
*DMR6*	Solyc03g080190	2	1	3	0.5	1.5	7	19	7	19	434	775	217.0	387.5
*DND1*	Solyc02g088560	3	2	2	0.7	0.7	16	38	18	46	410	806	136.7	268.7
*MLO1*	Solyc04g049090	13	6	16	0.5	1.2	67	120	60	121	5309	7787	408.4	599.0
*CPR5*	Solyc04g054170	1	0	0	0.0	0.0	2	6	6	9	653	873	653.0	873.0
*BIK1*	Solyc10g084770	51	18	41	0.4	0.8	237	452	272	500	12,789	21,376	250.8	419.1
*SR1*	Solyc01g105230	1	0	2	0.0	2.0	9	24	4	15	89	257	89.0	257.0
**Total**	-	125	58	119	-	-	715	1326	810	1454	33,710	54,378	-	-
**Average**	-	13	6	12	0.5	1.0	72	133	81	145	3371	5438	269.5	429.7

**Table 2 plants-12-02289-t002:** Detailed statistics on the allelic richness of the tomato genotypes (BIG, from Lin et al. 2014 [38]) considering the high-impact SNPs in the whole gene dataset and in the selected S-genes.

				High-Impact SNPs			High-Impact SNPs in S-Genes		
Genotype	Name	TGRC/PI-CGN/EA	Categories	Total	Hom.	Heteroz.	Total	Hom.	Heteroz.
TS-214	Panama	-/-/-	Landrace	620	569	51	7	6	1
TS-074	N 739	-/-/-	Fresh market	647	587	60	5	5	0
TS-186	Rowpac	LA3214/-/-	Modern processing	445	423	22	5	5	0
TS-007	Micro-Tom	LA3911/-/-	Modern fresh market	901	724	177	4	4	0
TS-224	Guayaquil	LA0410/PI 258474/-	Landrace/Latin American cultivar	779	767	12	4	4	0
TS-296	Droplet	-/-/-	-	719	668	51	4	4	0
TS-409	-	-/PI124161/-	Landrace	1526	1263	263	4	4	0
TS-003	M-82	LA3475/-/-	Modern processing	515	424	91	3	3	0
TS-004	Hawaii 7998	LA3856/-/-	Inbreed line	692	606	86	3	3	0
TS-011	KR2	-/-/-	Modern fresh market	565	392	173	5	3	2
TS-135	Hacienda Rosario	LA0466/PI 258469/-	Landrace/Latin American cultivar	334	301	33	3	3	0
TS-150	Tarapoto	LA2285/-/-	Landrace/Latin American cultivar	352	326	26	3	3	0
TS-190	Santa Chiara	-/-/-	Cultivar	437	366	71	3	3	0
TS-277	Hunt100	LA3144/-/-	Modern processing	266	236	30	3	3	0
TS-005	Edkawi	LA2711/-/-	Vintage fresh market	191	116	75	3	2	1
TS-012	yoku improvement	-/-/-	Modern fresh market	505	400	105	4	2	2
TS-078	-	-/-/EA02895	Processing tomato	300	273	27	2	2	0
TS-089	-	-/-/EA01185	Processing tomato	457	371	86	3	2	1
TS-090	-	-/-/EA02753	Cocktail tomato	368	286	82	2	2	0
TS-108	Puno I	-/-/EA01989	Processing tomato	334	312	22	2	2	0
TS-121	NC EBR-6	LA3846/-/-	Modern fresh market	267	225	42	2	2	0
TS-122	Rutgers	LA1090/-/-	Vintage fresh market	70	58	12	2	2	0
TS-127	Hacienda Calera	LA0113/-/-	Landrace/Latin American cultivar	1589	886	703	3	2	1
TS-143	Florida 7547	LA4025/-/-	Modern fresh market	182	163	19	2	2	0
TS-147	-	-/-/-	-	482	404	78	2	2	0
TS-171	UC-82	LA1706/-/-	Modern processing	334	305	29	3	2	1
TS-204	Florida 7060	LA3840/-/-	Modern fresh market	247	202	45	2	2	0
TS-220	Barnaulski Konservnyi	-/-/-	Cultivar	535	455	80	2	2	0
TS-225	-	-/PI330336/EA05747	Processing tomato	172	108	64	3	2	1
TS-226	Microtom	-/-/-	Cultivar	436	400	36	3	2	1
TS-228	M-82	-/-/-	Cultivar	398	369	29	2	2	0
TS-234	-	-/-/EA01371	Processing tomato	234	219	15	2	2	0
TS-237	Platense	LA3243/-/-	Vintage fresh market	190	145	45	2	2	0
TS-245	-	-/-/EA03126	Processing tomato	314	248	66	4	2	2
TS-276	-	-/-/EA03650	Cocktail/processing tomato	160	124	36	3	2	1
TS-292	-	-/-/EA06902	Processing tomato	298	278	20	2	2	0
TS-002	Moneymaker	LA2706/-/-	Vintage fresh market	207	151	56	2	1	1
TS-008	E-6203	LA4024/-/-	Modern processing	380	302	78	4	1	3
TS-009	Ailsa Craig	LA2838A/-/-	Vintage fresh market	182	128	54	2	1	1
TS-041	-	-/-/EA02435	Cocktail tomato	262	218	44	1	1	0
TS-043	Moneymaker	-/-/EA00840	Fresh market	166	130	36	1	1	0
TS-045	-	-/PI303718/EA05578	Processing tomato	198	176	22	1	1	0
TS-047	-	-/-/EA01960	Processing tomato	144	125	19	1	1	0
TS-049	Earliana	LA3238/-/-	Vintage processing	149	139	10	1	1	0
TS-051	-	-/-/-	-	127	100	27	1	1	0
TS-052	05-4126 (97-49-2)	-/-/-	Cultivar	328	281	47	2	1	1
TS-055	-	-/-/EA00448	-	176	117	59	1	1	0
TS-058	-	-/-/EA03577	Processing tomato	131	119	12	1	1	0
TS-059	-	-/-/EA02898	Processing tomato	690	516	174	1	1	0
TS-068	Chiclayo	LA0395/-/-	Latin American cultivar	1640	185	1455	9	1	8
TS-069	Huachinango	LA1459/-/-	Latin American cultivar	247	231	16	1	1	0
TS-073	Quarantino	-/-/-	-	126	105	21	1	1	0
TS-076	-	-/-/EA01230	Processing tomato	156	129	27	1	1	0
TS-081	-	-/-/EA02761	Processing tomato	182	155	27	1	1	0
TS-085	-	-/-/-	-	474	237	237	3	1	2
TS-086	-	-/-/EA01684	-	139	118	21	1	1	0
TS-095	Moneymaker	-/-/-	Fresh market	176	147	29	2	1	1
TS-100	-	-/-/EA03456	Processing	134	117	17	1	1	0
TS-112	-	-/-/EA03083	Processing tomato	175	148	27	1	1	0
TS-115	-	-/-/EA03426	Processing tomato	243	222	21	1	1	0
TS-117	Scatolone di bolsena	-/-/-	Landrace	214	104	110	1	1	0
TS-125	-	-/-/EA00422	Processing tomato	241	137	104	2	1	1
TS-128	Pearson	LA0012/-/-	Vintage processing	245	214	31	1	1	0
TS-132	Primabel	LA3903/-/-	Vintage fresh market	136	116	20	1	1	0
TS-133	Peto95-43	LA3528/-/-	Modern processing	307	264	43	1	1	0
TS-137	Spagnoletta	-/-/-	Landrace	305	136	169	1	1	0
TS-142	Roma	-/-/-	Vintage cultivar	136	122	14	2	1	1
TS-151	T-5	LA2399/-/-	Modern fresh market	625	529	96	2	1	1
TS-152	Santa Cruz B	LA1021/-/-	Landrace/Latin American cultivar	177	160	17	1	1	0
TS-155	Condine Red	LA0533/-/-	Vintage fresh market, monogenic	130	119	11	1	1	0
TS-157	-	-/-/EA03648	Processing tomato	121	104	17	1	1	0
TS-160	-	-/-/EA03533	Processing tomato	221	185	36	1	1	0
TS-163	Marmande	LA1504/-/-	Vintage fresh market	129	114	15	1	1	0
TS-166	Piura	LA0404/-/-	Landrace/Latin American cultivar	178	163	15	2	1	1
TS-167	Tegucigalpa	LA0147/-/-	Landrace/Latin American cultivar	158	135	23	1	1	0
TS-168	-	-/-/-	Landrace	337	256	81	1	1	0
TS-174	-	-/-/EA00304	Processing tomato	212	191	21	1	1	0
TS-176	-	-/-/EA02669	Processing tomato	197	190	7	1	1	0
TS-177	-	-/-/EA01155	Processing tomato	127	108	19	1	1	0
TS-180	-	-/-/EA02728	Processing tomato	116	82	34	1	1	0
TS-183	-	-/-/EA02764	Processing tomato	154	133	21	1	1	0
TS-184	Tarapoto	LA2283/-/-	-	338	225	113	2	1	1
TS-193	Pantano dArdea	-/-/-	Landrace	170	121	49	1	1	0
TS-194	-	-/-/-	-	167	143	24	1	1	0
TS-197	Libanese	-/-/-	Landrace	165	122	43	1	1	0
TS-198	-	-/-/EA00512	-	153	129	24	1	1	0
TS-200	Hot set	LA3320/-/-	Cultivar	187	135	52	1	1	0
TS-203	Bell pepper-like	-/-/-	Landrace	177	110	67	1	1	0
TS-206	Prince Borghese	LA0089/-/-	Vintage fresh market	26	22	4	1	1	0
TS-211	NC 84173	LA4354/-/-	Modern fresh market	425	366	59	1	1	0
TS-215	Vrbikanske Nizke	-/-/-	Cultivar	183	126	57	2	1	1
TS-235	-	-/-/EA00892	Processing tomato	46	44	2	1	1	0
TS-239	NC EBR-5	LA3845/-/-	Modern fresh market	126	109	17	2	1	1
TS-242	Ayacucho	LA0134C/-/-	Latin American cultivar	530	332	198	1	1	0
TS-251	-	-/PI647249/EA04001	-	150	128	22	1	1	0
TS-256	-	LA2260/0/EA00744	Latin American cultivar	477	415	62	1	1	0
TS-261	-	LA1511/-/EA01444	Wild species	246	145	101	2	1	1
TS-263	Rio Grande	LA3343/-/-	Processing tomato	213	183	30	1	1	0
TS-264	King Humbert #1	LA0025/-/-	Vintage fresh market	134	119	15	1	1	0
TS-268	-	-/-/EA01915	Cultivar	147	130	17	1	1	0
TS-274	-	-/-/EA03613	Cocktail/processing tomato	266	241	25	1	1	0
TS-278	Early Santa Clara	LA0517/-/-	Vintage processing	207	187	20	1	1	0
TS-400	-	-/-/-	Inbred line	453	398	55	1	1	0

**Table 3 plants-12-02289-t003:** Disease assay with *O. neolycopersici* performed on four varieties and a control variety (Moneymaker). The disease score (DS) values reported here were compared with the ones derived from the controls. Statistical differences among the varieties/control were analyzed with a two-tailed *t* test (*p* < 0.05).

Variety	Code	Type	DS (0–3)	Std. Error	n	*p*-Value	Reduction (%)	Class
VF-36	TS-1	control	3.00	0	20	-	-	a
Money Maker	TS-2	assayed	2.96	0.03	28	0.326189	1.2%	a
Droplet	TS-296	assayed	2.87	0.09	15	0.164318	4.4%	a
M-82	TS-003	assayed	2.42	0.14	33	0.000367	19.2%	b
Puno-I	TS-108	assayed	2.67	0.11	21	0.004900	11.1%	b

## Data Availability

The sequencing data used in this study are openly available in the NCBI database (SRA/SRP045767).

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
