# Peer review of "Genomic Analysis Highlights Putative Defective Susceptibility Genes in Tomato Germplasm"

_plants, 2023, doi:10.3390/plants12122289_

Round 1
Reviewer 1 Report
Summary:
This study identified thousands of potential susceptibility alleles that could be targeted for disease resistance breeding. Ten susceptibility gene families were surveyed over 360 tomato accessions in a publicly available data set. Ten of the S gene SNPs were verified with Sanger sequencing. Five of the tomato accessions were screened for powdery mildew resistance. Ten potential sgDNA guides were designed for these SNPs. This work highlights the power of publicly available data for additional analyses and more importantly, the trove of potentially useful alleles present already in the tomato breeding pool. These genes, if shown to reduce disease susceptibility, could prove to be durable sources of resistance against pathogens.
Major Comments:
The biggest concern we have with this study is that the 5 accession pilot disease screen doesn’t strongly support the conclusion that the defective S alleles were the source of the reduced susceptibility. An alternative hypothesis to the tested accessions having defective S alleles is that the accessions have R genes. One of the tested accessions, Droplet, was said to have resistance to Verticillium wilt in its varietal release (ref 1). Then a progeny of Droplet (Prairie Schooner) was reported to have Ve, I, and Pto resistance genes to fungal pathogens (ref 2). It could be that the R genes present in Droplet have some effectiveness against powdery mildew. We only checked one accession for R genes for an example.
Compelling evidence would be if CRISPR edited mutants did show reduced disease compared to their wild type progenitor. To show that any of the S gene alleles mediate resistance, the authors need to use their sgRNA CRISPR guides to edit the functional S genes. Disease assays should be run on the wild type and edited plants, and t-tests performed. Another common study for providing strong support of an allele’s effect would be a genome wide association study (GWAS) over the 360 individuals, or at least a large subsample. The authors expressed that obtaining the germplasm is difficult at their institute, so a GWAS would likely be unfeasible. Perhaps the two first authors, as they are students who might relish the training opportunity, could spend time as visiting scientists at a collaborating institute with access to the germplasm that could run a disease screen.
Phenotypic information produced from the screens themselves is missing, was expecting some type of figure to show results as well as a detailed table with 0-3 ratings and experimental design indicating replication etc.
As the study is now, there is not sufficient evidence that the S alleles impact susceptibility. Therefore, these claims should be removed from the paper:
-
Line 27: “The current results provide a resource for tomato genomic-assisted breeding programs as well as tailored gene editing approaches for disease resistance”
-
Thus, our results provide a valuable resource for plant genetics, with potential applications in genomic-assisted breeding programs for resistance to biotic stresses.”
Testing CRISPR generated mutants would have provided strong evidence that the S alleles affect disease susceptibility so it is misleading that the authors refer to their S alleles as mutants in these cases:
-
Line 24: “Three homozygous S-genes mutants were infected with Oidium neolycopersici, and two highlighted a significantly reduced susceptibility to the fungus”
-
Line 113: “Homozygous S-gene 113 mutants were assayed for resistance to O. neolycopersici, of which two highlighted a reduced susceptibility.”
-
Line 337: “Furthermore, we validated the identified SNPs through Sanger sequencing and found that some homozygous mutants exhibited a significantly reduced susceptibility to powdery mildew.”
Recommendations to improve the study:
If developing CRISPR mutants or conducting a GWAS are implausible, then this study could be improved by including a few additional bioinformatics analyses -or- phrasing more in line with what has been done, a survey of gene family clusters of previously identified susceptibility genes. The first that we would recommend is use the same resequencing data, but 1) update the vcf with the newest tomato genome assembly, 2) compare S allele haplotypes (collection of SNPs within or near each gene) and/or 3) compare S allele haplotypes between the distinct domestication groups.
-
The current tomato reference genome is SL 5.0. The WGS data used to generate these vcf files is available on NCBI (https://www.ncbi.nlm.nih.gov/sra/SRP045767). New vcf files with the current genome assembly could easily be generated. Updating the vcf files with the modern reference genome (SL v5.0) would likely improve results as the vcf used in the study was aligned to a reference genome that is over 9 years old. Reference genome quality has exponentially increased in that time.
-
The vcf files could be used to generate haplotypes around S genes. The haplotypes could simply be the collection of SNPs in and within a few kb of each gene. Generating haplotypes of SNPs would help support that these alleles are unique and not just artifacts of genotype calling. It could be useful for breeders to know which S alleles are shared among cultivars.
-
The comparison of S alleles specific to different groups (PIM, CER and BIG) could be made to show genetic bottlenecks. The hypothesis here would be that PIM would have the most diversity in S alleles, then CER with a moderate number, and BIG would have the least.
A second study we might recommend is a comparative genomics study. The authors could directly compare S alleles from the tomato pan-genome (ref 3) assemblies to identify novel alleles. The genome assemblies and their annotations are publicly available.
References we used:
-
Ref 1) Honma, S. and H.M. Murakishi. (1971). Droplet—A new fresh-market tomato. Michigan State Univ. Res. Rpt. 138.
-
Ref 2) Summers, W. L. (1996). `Prairie Schooner' Inbred Cherry Tomato, HortScience HortSci, 31(2), 291-291. Retrieved from https://doi.org/10.21273/HORTSCI.31.2.291
-
Ref 3) Zhou, Y., Zhang, Z., Bao, Z. et al. Graph pangenome captures missing heritability and empowers tomato breeding. Nature 606, 527–534 (2022). https://doi.org/10.1038/s41586-022-04808-9
Minor Comments:
Writing:
-
Many pathogens and pests are discussed in this paper. It would help the readers quite a bit if the authors presented the disease common name with corresponding species and then referred to that pest/pathogen by the only one name for the rest of the paper.
-
It would also help the readability if abbreviations are avoided where possible. Gene names with abbreviations are fine, SNPs is fine, but disease and pathogen should be used instead of abbreviations. Other abbreviations to drop completely are EFSA, NGTs, and HoSU.
-
It would help to include a quick summary of the plant germplasm used in the introduction or beginning of results (if methods come after results, as they did in the copy of the paper we received). We were very confused about the BIG and ALL designations for most of the paper. Consider explaining why those two were considered.
Scientific:
-
BIK1 and all of the other receptor-like kinases are canonically part of the resistance pathways. A stronger argument for including it as an S gene should be made.
-
Known R genes in the plants that were disease assayed should be disclosed, as R genes could be responsible for the resistance observed.
-
Which genome assembly was used to generate the vcf file? Figure 1 mentions SL v2.4, but the methods mention v2.5.
-
Line 112: “Identified mutation were screened to assess their impact on protein structure.”
-
only predicted SNP effects were analyzed. Protein structures were not analyzed.
-
Line 132-133, ‘analyzed by bioinformatics’ isn’t very meaningful here. Tell us what kind of bioinformatic analyses were used.
Tables and figures:
-
Figure 1 - List full number of tomato genes or simply list 34K; that number is discrepant from listed in prose (L133 indicated as 30K). What is it really in that genome version?
-
Figure 2 - Y-axis labels are needed. Use separated bar charts for both a and b to keep consistent, stacked bar chart is a little harder to read.
-
It’s not clear what the PCA in Figure 3 is trying to show or what data was used to generate it. All of the SNPs? Only SNPs in S genes?
-
Lines 198-200 are confusing because Table S5 doesn’t indicate which SNPs the dozen accessions carry.
-
Lines 193-195 mention that some SNPs are conserved across many accessions, but the data presented in S3 and S6 do not clearly support this statement.
-
Lines 189-190 and 219-221 present the same information:
-
189-190: “Homozygous SNPs/indels - The number of genotypes with two SNPs was 174 (whole dataset) and 76 (BIG tomatoes), while those with three or more SNPs were 114 and 14 (Table 2, Figure 4), respectively.”
-
219-221: “If two SNPs are considered, the number of genotypes increases to 174 (ALL) and 76 (BIG), and if three SNPs are considered, the number of genotypes decreases to 114 (ALL) and 14 (BIG). “
Author Response
Comments and Suggestions for Authors
Summary:
This study identified thousands of potential susceptibility alleles that could be targeted for disease resistance breeding. Ten susceptibility gene families were surveyed over 360 tomato accessions in a publicly available data set. Ten of the S gene SNPs were verified with Sanger sequencing. Five of the tomato accessions were screened for powdery mildew resistance. Ten potential sgDNA guides were designed for these SNPs. This work highlights the power of publicly available data for additional analyses and more importantly, the trove of potentially useful alleles present already in the tomato breeding pool. These genes, if shown to reduce disease susceptibility, could prove to be durable sources of resistance against pathogens.
Authors: We thank the Reviewer for this valuable comment and have made use of the last part (underlined) at the end of our conclusions (sligthly revised), with which we hope that the Reviewer agrees.
Major Comments:
REVIEWER1: The biggest concern we have with this study is that the 5 accession pilot disease screen doesn’t strongly support the conclusion that the defective S alleles were the source of the reduced susceptibility. An alternative hypothesis to the tested accessions having defective S alleles is that the accessions have R genes. One of the tested accessions, Droplet, was said to have resistance to Verticillium wilt in its varietal release (ref 1). Then a progeny of Droplet (Prairie Schooner) was reported to have Ve, I, and Pto resistance genes to fungal pathogens (ref 2). It could be that the R genes present in Droplet have some effectiveness against powdery mildew. We only checked one accession for R genes for an example.
Authors: We thank the reviewer for the comments. The presence of R genes in the tested accessions might be a possible hypothesis. Indeed, some R genes may provide resistance to a relatively wide range of pathogens, but this is not their common feature. Differently, S-genes facilitate the interaction between plant and pathogens and their disabling provide a more broad-spectrum and durable type of resistance and has been defined as horizontal (van Schie et al 2014, Review of Phytopathology, 52(1),551-581; Pavan et al 2009, Molecular Breeding 25 (1). The goal of our study was to identify defective S genes and their putative role in causing tolerance/resistance to pathogens. The contemporary presence of R genes was out of our scope as well as the assessment of side effects caused by their mutation, which would require a one-by-one assessment of their usefulness for application.
REVIEWER1: Compelling evidence would be if CRISPR edited mutants did show reduced disease compared to their wild type progenitor. To show that any of the S gene alleles mediate resistance, the authors need to use their sgRNA CRISPR guides to edit the functional S genes. Disease assays should be run on the wild type and edited plants, and t-tests performed. Another common study for providing strong support of an allele’s effect would be a genome wide association study (GWAS) over the 360 individuals, or at least a large subsample. The authors expressed that obtaining the germplasm is difficult at their institute, so a GWAS would likely be unfeasible. Perhaps the two first authors, as they are students who might relish the training opportunity, could spend time as visiting scientists at a collaborating institute with access to the germplasm that could run a disease screen.
Authors: Indeed, both GWAS and CRISPR approaches might provide more robust evidence but this was out of the scope of our present manuscript. The two suggested strategies would require a lot of time and will be the subject of future work. Nevertheless, we have updated the manuscript according to the reviewer's suggestions (adding 4 new supplementary figures and 1 Supplementary table, see below).
Moreover, thanks to the referee’s comment, we propose to change the title from:
“Genomic Analysis Reveals Defective Susceptibility Genes in Tomato Germplasm”, to:
“Genomic analysis highlights putative defective susceptibility genes in tomato germplasm
We have also modified, accordingly, indicating the defective alleles as “putative”:
- in Results and Discussion (line 130): “This work represents the first genomic assessment a of S-genes and putative defective alleles in the Solanaceae family.”
- in the Conclusions (line 366-367): “In this study, we conducted a comprehensive genomic survey of various tomato genotypes to identify putative defective alleles of susceptibility genes
REVIEWER1: Phenotypic information produced from the screens themselves is missing, was expecting some type of figure to show results as well as a detailed table with 0-3 ratings and experimental design indicating replication etc.
Authors: Thank you for the comment. We have prepared a new supplementary table (S7) with raw data and a figure (Figure S3) showing some images of the infected plants following the pathogen infection. Details related to the randomisation of the experimental scheme were added in the manuscript (Line: 360).
REVIEWER1: As the study is now, there is not sufficient evidence that the S alleles impact susceptibility. Therefore, these claims should be removed from the paper:
- Line 27: “The current results provide a resource for tomato genomic-assisted breeding programs as well as tailored gene editing approaches for disease resistance”
- Thus, our results provide a valuable resource for plant genetics, with potential applications in genomic-assisted breeding programs for resistance to biotic stresses.”
Authors: Thanks to the referee for the suggestions. We have removed the two statements accordingly.
REVIEWER1: Testing CRISPR generated mutants would have provided strong evidence that the S alleles affect disease susceptibility so it is misleading that the authors refer to their S alleles as mutants in these cases:
Line 24: “Three homozygous S-genes mutants were infected with Oidium neolycopersici, and two highlighted a significantly reduced susceptibility to the fungus”
Line 113: “Homozygous S-gene mutants were assayed for resistance to O. neolycopersici, of which two highlighted a reduced susceptibility.”
Line 337: “Furthermore, we validated the identified SNPs through Sanger sequencing and found that some homozygous mutants exhibited a significantly reduced susceptibility to powdery mildew.”
Authors: Thanks to the referee for the suggestions. We have modified the three statements accordingly, as follows:
Line 24: “Three genotypes carrying high-impact homozygous SNPs in S-genes were infected with Oidium neolycopersici, and two highlighted a significantly reduced susceptibility to the fungus”
Line 113: “Genotypes carrying high-impact homozygous SNPs in S-genes were assayed for resistance to O. neolycopersici, of which two highlighted a reduced susceptibility.”
Line 337: “Furthermore, we validated the identified SNPs through Sanger sequencing and found out that two genotypes carrying high-impact homozygous SNPs in S-genes exhibited a significantly reduced susceptibility to powdery mildew.”
Recommendations to improve the study:
REVIEWER1: If developing CRISPR mutants or conducting a GWAS are implausible, then this study could be improved by including a few additional bioinformatics analyses -or- phrasing more in line with what has been done, a survey of gene family clusters of previously identified susceptibility genes. The first that we would recommend is use the same resequencing data, but 1) update the vcf with the newest tomato genome assembly, 2) compare S allele haplotypes (collection of SNPs within or near each gene) and/or 3) compare S allele haplotypes between the distinct domestication groups.
Authors: Thanks to the referee. We agree and we have implemented the manuscript according to the suggestions as reported below in the paragraphs that follow.
REVIEWER1: The current tomato reference genome is SL 5.0. The WGS data used to generate these vcf files is available on NCBI (https://www.ncbi.nlm.nih.gov/sra/SRP045767). New vcf files with the current genome assembly could easily be generated. Updating the vcf files with the modern reference genome (SL v5.0) would likely improve results as the vcf used in the study was aligned to a reference genome that is over 9 years old. Reference genome quality has exponentially increased in that time.
Authors: We agree with the Reviewer and we have implemented the SNPs coordinates with the new tomato reference genome (lift-over of the coordinates of SL5.0); table S6 was modified consequently.
REVIEWER1: The vcf files could be used to generate haplotypes around S genes. The haplotypes could simply be the collection of SNPs in and within a few kb of each gene. Generating haplotypes of SNPs would help support that these alleles are unique and not just artifacts of genotype calling. It could be useful for breeders to know which S alleles are shared among cultivars.
Authors: We have implemented the haplotype analysis on conserved SNPs, which shows they are not artifacts due to genotype calling. We have implemented it in material and methods (lines 335-337) and in results (167-169) sections. A new supplementary figure (Figure S1) has been produced.
REVIEWER1: The comparison of S alleles specific to different groups (PIM, CER and BIG) could be made to show genetic bottlenecks. The hypothesis here would be that PIM would have the most diversity in S alleles, then CER with a moderate number, and BIG would have the least.
Authors: We have added a diversity analysis among different groups using the Pi diversity index, and implemented material and methods (lines 316-319) and results (198-208). We have also added a new Figure S2.
REVIEWER1: A second study we might recommend is a comparative genomics study. The authors could directly compare S alleles from the tomato pan-genome (ref 3) assemblies to identify novel alleles. The genome assemblies and their annotations are publicly available.
Authors: Thank you for your suggestion. We agree that this analysis would be of interest, and the subject of future studies, as it would require very extensive work that goes beyond the scope of our current manuscript, whose goal is to perform a survey of gene family clusters of previously identified susceptibility genes.
Minor Comments:
Writing:
REVIEWER1: Many pathogens and pests are discussed in this paper. It would help the readers quite a bit if the authors presented the disease common name with corresponding species and then referred to that pest/pathogen by the only one name for the rest of the paper.
Authors:. Results were mainly focused on Oidium neolycopersici, the causal agent of powdery mildew; thus, after introducing it initially, we now refer to it as O. neolycopersici throughout the rest of the paper. We have also eliminated acronyms (e.g.: LB) throughout the manuscript.
REVIEWER1: It would also help the readability if abbreviations are avoided where possible. Gene names with abbreviations are fine, SNPs is fine, but disease and pathogen should be used instead of abbreviations. Other abbreviations to drop completely are EFSA, NGTs, and HoSU.
Authors: We have eliminated abbreviations for diseases (e.g.: LB, PM). We have also eliminated NGTs and HOSU, as abbreviations. We did not eliminate the EFSA abbreviation, since it is the common name used in Europe for European Food Safety Authority.
REVIEWER1: It would help to include a quick summary of the plant germplasm used in the introduction or beginning of results (if methods come after results, as they did in the copy of the paper we received). We were very confused about the BIG and ALL designations for most of the paper. Consider explaining why those two were considered.
Authors: We have now inserted at the beginning of the results a quick summary of the plant germplasm used in the work. BIG are a collection of 168 big-fruited S. lycopersicum accessions, while the whole collection of 360 genotypes was referred to ALL (we proposed the same names of the original paper (Lin et al).
Scientific:
REVIEWER1: BIK1 and all of the other receptor-like kinases are canonically part of the resistance pathways. A stronger argument for including it as an S gene should be made
Authors: Many studies (cited), including Schie et al. (2014), Pavan et al. (2015), Huibers et al. (2013), and Sun et al. (2016), have identified BIK1 as a key player conferring broad-spectrum resistance against various pathogens. In general, BIK1 is a negative regulator of immune signaling as a PTI suppressor involved in the direct or indirect suppression of SA-mediated defence pathways (Schie et al 2014). BIK has been classified as an S-gene that enhances biotroph resistance but reduces necrotroph resistance.
REVIEWER1: Known R genes in the plants that were disease assayed should be disclosed, as R genes could be responsible for the resistance observed.
Authors: To our knowledge, information about R genes present in the genotype used is scarce. R-genes contained in ‘Prairie Schooner’ variety (Droplet x Ontario 7710), Ve, I, and Pto are not inherited from Droplet, but from the other accession used, Ontario 7710 (Can. J. Plant Sci. 63: 1107-1109 (Oct. 1983)). Even if Farthest North, the variety that likely gave Pto gene to Ontario 7710 was used also in Droplet development, there is no information for Pto gene present in Droplet. In VF-36 are contained two R-gene: N, effective against tobacco mosaic virus, and Ve, effective against Verticillium (doi: 10.1111/j.1365-313X.2005.02380.x). It is also known that M-82 contains two R-gene: Fol2 and I, effective against Fusarium oxysporum (https://doi.org/10.3390/genes12111673). No information is reported for R-gene in Money Maker and Puno-I backgrounds.
In conclusion, literature does not report R-gene for resistance against Oidium neolycopersici in the genotypes used and it is unlikely that other R genes could be responsible for the resistance observed. See the answer to the first question.
REVIEWER1: Which genome assembly was used to generate the vcf file? Figure 1 mentions SL v2.4, but the methods mention v2.5.
Authors: Thank you for the comment, it was a typo. The correct citation is SL2.5. We have corrected the figure.
REVIEWER1: Line 112: “Identified mutation were screened to assess their impact on protein structure.” only predicted SNP effects were analyzed. Protein structures were not analyzed.
Authors: We have changed the phrase to: “Identified mutations were screened to assess their likely impact on protein functionality”.
REVIEWER1: Line 132-133, ‘analyzed by bioinformatics’ isn’t very meaningful here. Tell us what kind of bioinformatic analyses were used.
Authors: We have changed the phrase to: “were analyzed for SNP mining”
Tables and figures:
REVIEWER1: Figure 1 - List full number of tomato genes or simply list 34K; that number is discrepant from listed in prose (L133 indicated as 30K). What is it really in that genome version?
Authors: Thank you for the suggestion. We have now inserted the suggested number in figure 1 (34k) and the precise one at L133 (34,725 genes).
REVIEWER1: Figure 2 - Y-axis labels are needed. Use separated bar charts for both a and b to keep consistent, stacked bar chart is a little harder to read.
Authors: Thank you for the suggestion. We have now put the y-axis labels. We have also modified the a) bar chart, avoiding staked bars
REVIEWER1: It’s not clear what the PCA in Figure 3 is trying to show or what data was used to generate it. All of the SNPs? Only SNPs in S genes?
Authors: The PCA plot in Figure 3 was created to display the diversity of germplasm varieties based on their SNPs, with a specific focus on those carrying numerous deleterious SNPs in S-genes. The SNPs used to generate the plot included total SNPs filtered for a maximum of 20% missing data for each SNP.
REVIEWER1: Lines 198-200 are confusing because Table S5 doesn’t indicate which SNPs the dozen accessions carry.
Authors: Thank you for the comment. It was a typo. The correct citation is Table S6. We have corrected it in the manuscript.
REVIEWER1: Lines 193-195 mention that some SNPs are conserved across many accessions, but the data presented in S3 and S6 do not clearly support this statement.
Authors: Thank you for the comment. We have removed the reference to Table S3 (wrong) and confirmed that in Table s6 some SNPs resulted as redundant among genotypes (eg. SNP 852558 in ch3 in TS127 and TS 409; SNP 50610274 in ch2 in TS135, TS150 and TS 89)
REVIEWER1: Lines 189-190 and 219-221 present the same information:
189-190: “Homozygous SNPs/indels - The number of genotypes with two SNPs was 174 (whole dataset) and 76 (BIG tomatoes), while those with three or more SNPs were 114 and 14 (Table 2, Figure 4), respectively.”
219-221: “If two SNPs are considered, the number of genotypes increases to 174 (ALL) and 76 (BIG), and if three SNPs are considered, the number of genotypes decreases to 114 (ALL) and 14 (BIG). “
Authors: Thank you for the comment. We have now corrected the two phrases as follows:
- Lines 189-190: “The number of genotypes with two SNPs was 174 (whole dataset) and 36 (BIG toma-toes), while those with three or more SNPs were 114 and 14 (Table 2, Figure 4), respectively. “
- Lines 219-221: “If two SNPs are considered, the number of genotypes was 89 (ALL) and 10 (BIG), while if three SNPs are considered, the number of genotypes decreases to 54 (ALL) and 3 (BIG).”
Reviewer 2 Report
In the abstract, define HSP in line 10.
In Matherial and Metods, add a description and origin of the E42 genotype.
In page 10, the authors claim “Since the phylogenetic analysis evidenced the relationship of E42 genotype with the heat-tolerant Solanum pimpinellifolium wild species”. In the phylogenetic tree of chromosome 1 (Figure 2) E42 genotype does not group with Solanum pimpinellifolium. The authors need to clarify their claim.
Author Response
Reviewer2: In the abstract, define HSP in line 10.
Authors: We have eliminated acronyms in the abstract, as well as throughout the manuscript.
Reviewer2: In Matherial and Metods, add a description and origin of the E42 genotype.
Authors: The genotypes are now described at the beginning of the results, as well as in the Materials and Methods section and Table S2.
Reviewer2: In page 10, the authors claim “Since the phylogenetic analysis evidenced the relationship of E42 genotype with the heat-tolerant Solanum pimpinellifolium wild species”. In the phylogenetic tree of chromosome 1 (Figure 2) E42 genotype does not group with Solanum pimpinellifolium. The authors need to clarify their claim.
Authors: The phylogenetic analysis (PCA) was conceived to not include S. pimpinellifolium as it focused solely on "BIG” tomatoes.